# AI for Open Science: A Multi-Agent Perspective for Ethically Translating Data to Knowledge

## Abstract

AI for Science (AI4Science), particularly in the form of self-driving labs, has the potential to sideline human involvement and hinder scientific discovery within the broader community. While prior research has focused on ensuring the responsible deployment of AI applications, enhancing security, and ensuring interpretability, we also propose that promoting openness in AI4Science discoveries should be carefully considered. In this paper, we introduce the concept of AI for Open Science (AI4OS) as a multi-agent extension of AI4Science with the core principle of maximizing open knowledge translation throughout the scientific enterprise rather than a single organizational unit. We use the established principles of Knowledge Discovery and Data Mining (KDD) to formalize a language around AI4OS. We then discuss three principle stages of knowledge translation embedded in AI4Science systems and detail specific points where openness can be applied to yield an AI4OS alternative. Lastly, we formulate a theoretical metric to assess AI4OS with a supporting ethical argument highlighting its importance. Our goal is that by drawing attention to AI4OS we can ensure the natural consequence of AI4Science (e.g., self-driving labs) is a benefit not only for its developers but for society as a whole.

## 1 Introduction

The mission of science is the robust and efficient discovery and translation of knowledge. However, this process has historically been inefficient due to the inherently disjoint nature of the enterprise formed of multiple agents acting asynchronously with varied levels of collaboration (if at all). Yet, this asynchronous and disjoint behavior is natural as it is unrealistic for any single agent to be competent at all aspects of the knowledge discovery and translation processes, i.e., the processes by which raw data is collected via experimentation, mined for patterns to develop hypotheses, and tested to validate knowledge assertions that are disseminated via publication. However, with recent advances in the field of AI, particularly in deep learning, it is becoming increasingly clear that AI will transform these processes and usher in this new field of AI for Science (AI4Science) (1) with systems capable of being end-to-end experts in knowledge translation and discovery.

While this transformation is revolutionary, we contend that AI4Science has the potential to exacerbate issues, both ethical and practical, regarding effective collaboration and knowledge dissemination within the scientific community. For instance, self-driving labs, being a natural manifestation of AI4Science, pose to have the remarkable capacity to automate knowledge discovery, but what happens when these systems are closed? Further, how will we mitigate misaligned incentives and motivations that limit the dissemination of derived findings? In essence, what of openness in AI for Science?

It is critical we address these questions before such systems arrive. To that end, we present an extension of AI4Science under a collaborative multi-agent framing of Open Science (2), namely AI for Open Science (AI4OS). We represent AI4OS as a Multi-agent Discovery Support System

Submitted to NeurIPS 2023 AI for Science Workshop, Attention Track.

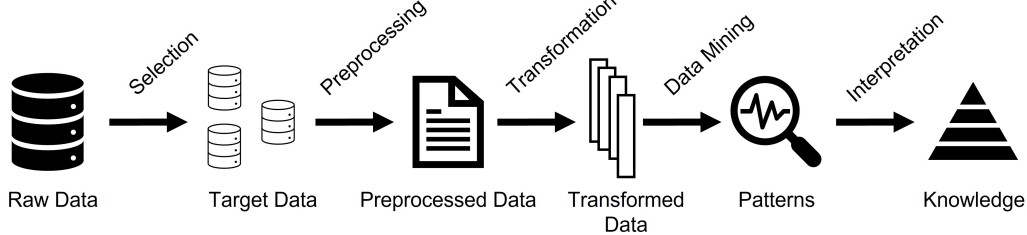

Figure 1: An Overview of the KDD process.

(MaDiSS) with the explicit intent of maximizing discovery for all stakeholders, transcending the limitations of isolated, self-serving systems that could arise in AI4Science implementations agnostic to openness.

While definitions of Open Science vary, we adopt that Open Science is "a collaborative culture facilitated by technology, promoting the open exchange of *data*, *information*, and *knowledge* among the scientific community and the general public, ultimately expediting scientific research and enhancing comprehension" (2). Crystallizing this sentiment with established definitions of *data*, *information*, and *knowledge* from the Knowledge Discovery and Data Mining (KDD) (3; 4) community, we develop a formal language to characterize each stage of the knowledge translation process. Leveraging this language, we derive a theoretical optimization metric for openness to build an ethical argument supporting AI4OS. Lastly, we use MaDiSS to offer recommendations for promoting openness in self-driving labs and other AI-driven systems.

**Contributions** This work makes three primary contributions:

1. Introduce a formal language for discussing issues of openness in AI4Science,

2. Develop a corresponding openness metric,

3. Construct a robust ethical argument in support of this framing.

## 2 Background and Related Work

### 2.1 The Knowledge Discovery and Data Mining Process

KDD provides useful and precise definitions of *data*, *information*, and *knowledge*, central to our definition of Open Science. The KDD literature (4; 5) defines *data* as a set of examples collected using experimentation and a *pattern* as an expression in some language describing a subset of data exemplars that is shorter than the enumeration of the entire dataset. We extend these definitions to include *information* as a set of these patterns that yield *knowledge*, a label placed on this information. The KDD process can be summarized in the following steps and is depicted in Figure 1.

1. Learning the application domain and acquiring prior knowledge necessary for the goals of the application,

2. Creating a target dataset which includes the selection or filtering over a subset of features or samples,

3. Data cleaning and preprocessing,

4. Data reduction, projection, and transformation,

5. Choosing the function of data mining, i.e., the required task that will be solved such as summarization, classification, regression, etc.,

6. Selecting the data mining algorithm and appropriate parameters required to run the model on the desired data,

7. Data mining or searching for patterns of interest in a particular representation form such as clustering, dependency analysis, rules, etc.,

8. Interpretation of the generated patterns and translating the useful ones into terms understandable by the user.

As stated by Fayyad et al. (3), the KDD process is iterative and may contain loops between any two steps in the process. For our purposes, the main limitation of KDD is that it assumes a single-agent perspective. However, we argue that a single-agent perspective is insufficient to model Open Science as it is inherently collaborative among many disjoint agents. Rather, it is this multi-agent communication/collaboration that is crucial to a successful knowledge translation. Therefore, we should consider three additional elements of the knowledge translation process, which to the best of our knowledge, have not been formulated into a single coherent framework. These elements correspond to the successful communication of prior knowledge (1) from the experimenter to the data miner and/or knowledge interpreter, (2) from the data miner to the knowledge interpreter, and (3) the decisive review of all propagated knowledge by the interpreter.

## 2.2 Discovery Support Systems

As our representation of AI4OS is formulated as a MaDiSS, we believe a brief review of traditional Discovery Support Systems (DiSS) is warranted. The first mention of DiSS came in 1986 when Don R. Swanson developed a literature-based procedure for generating new hypotheses focused on biomedical information (6). The first hypothesis emerging from his process proposed that fish oil could be used to cure Raynaud's disease and was later tested as well as experimentally and clinically proved (6; 7; 8). This research launched the field of Literature-Based Discovery (LBD) (9; 10; 11; 12) to infer new and useful knowledge by logically connecting information fragments from disparate textual sources. As LBD has developed into an immense field, we point the reader to Gopalakrishnan et. al. (13) for a complete review of recent LBD approaches focused on the biomedical domain and for more general applications to Thilakaratne et al. (14) and Hue et al. (15). While LDB techniques can be leveraged in the context of our definition of MaDiSS, our framework and discussion are more general than any individual methodology for discovery. Further, these approaches do not focus on our main contribution which is the knowledge communication between and consideration of each agent in the translation process.

A prime example of a modern DiSS as well as a MaDiSS is the US National Institute of Health (NIH) National Center for Advancing Translational Science (NCATS) Translator program (16) which was founded with the mission to create a "system capable of integrating existing biomedical data sets and "translating" those data into insights to accelerate translational research, generate new hypotheses, and drive innovations in clinical care and drug discovery." Importantly, the NCATS program takes an Open Science approach where knowledge is shared and reasoned over by many different independent teams via API calls (17). These types of collaborative efforts come with their own set of ethical considerations as they pertain to Open Science (18; 19; 20; 21; 22), Open Data (23; 24; 25; 26) and the provenance of such systems (27; 28; 17) and our framing subsumes these issues as well.

## 3 A Formal Language of AI for Open Science

In this section, our aim is to introduce a formal language to characterize AI4OS. We propose that a useful way to frame AI4OS is through a MaDiSS that aligns with the principles of Open Science. The primary motivation is that such a framing should enable researchers to conduct ethical assessments of AI4Science-inspired systems in a unified manner.

Our formalism begins by considering the three primary roles in a MaDiSS (and KDD in general), which revolve around dataset curation, information extraction, and knowledge labeling. Recognizing that multiple agents can participate in each role, we define the knowledge required by each agent to fulfill their specific role and represent the transfer of this knowledge to downstream collaborators promoting an open exchange of insights and expertise within their respective roles. These knowledge priors and roles are as follows:

1. The knowledge base of the *experimenting agent*, generating the raw data associated with KDD Steps 1-4,

2. The knowledge base of the *data mining agent*, in their considerations and selections of the data mining algorithm of KDD Steps 5-7,

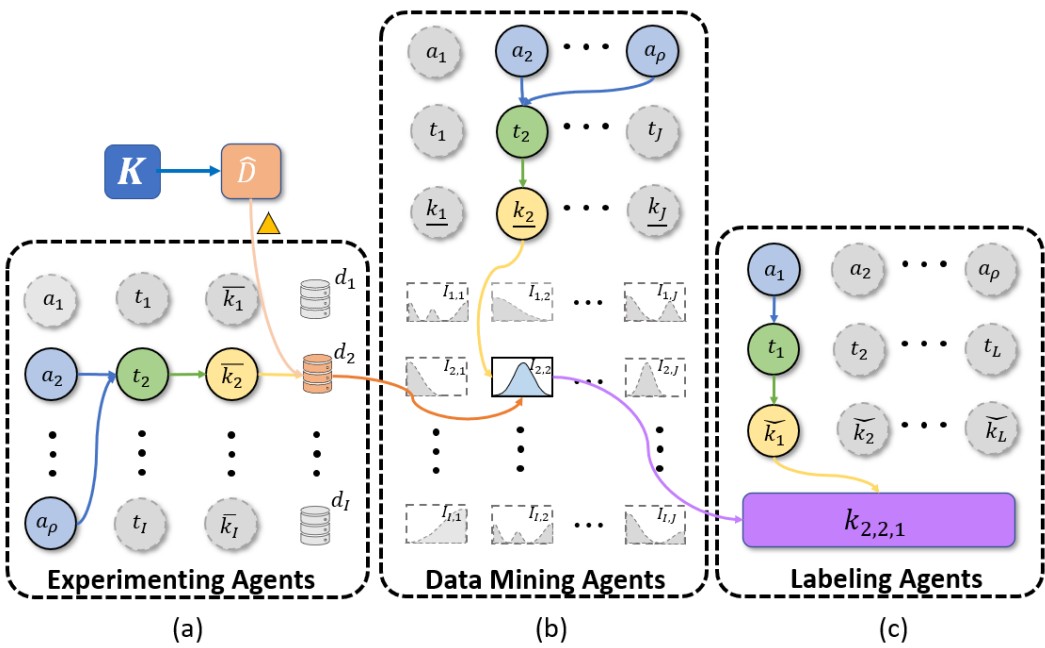

Figure 2: An overview of MaDiSS. The colored portions of the figure are illustrative of an arbitrary instantiation of the translation process. (a) the experimenting agent(s) responsible for constructing any particular dataset $d_i$ via experimentation. All $d_i$ represents some sub-population of the global ground truth data, $\hat{D}$ with some instrumentation error $\Delta$. (b) The data mining agent(s) responsible for the standard KDD process up until Step 7. (c) The labeling agent responsible for accepting/labeling new knowledge as derived from information presented by the data mining agent.

3. The knowledge base of the *labeling agent*, responsible for accepting, distilling and labeling the information patterns in KDD Step 8.

We quickly define some common notation that we will assume and exploit later. We assume the universe of knowledge consists of a (potentially infinite) set of true knowledge $K$ and a set of false knowledge (also potentially infinite) $K^c$. We let $A$ denote the set of all agents, human or otherwise, where $\rho = |A|$. Each agent has an assumed set of prior knowledge which is a subset of $K \cup K^c$. While operating over the three role types (experimenting, mining and labeling) previously introduced, we allow agents to form teams in which they can belong to multiple or no teams along each role axis. Therefore, its possible that the same set of agents forming the same team can perform each role in the pipeline. **This situation is analogous to a self-driving lab.** However, as in the real world, it is natural that an *experimenting agent* may have their data mined by many different *data mining agents*. Further, these mined patterns can be labeled by many different *labeling agents*. Since we are interested in the collaborative aspect of Open Science, MaDiSS provides a formalism for the translation of knowledge among teams. A visual depiction of MaDiSS is shown in Figure 2. In the following sections, we will outline each role more formally.

### 3.1 The Experimenting Agent

In the experimenting process (Figure 2.a), *experimenting agents* form sets of *experimenting teams*, $t_i \subseteq A$. Each team forms a team knowledge base, $\overline{k_i}$, assumed to rectify conflicts from $K^c$ and $K$ between agents as it pertains to the experimenting process. Data sets, $d_i \in D$, are collected by some means of experimentation which we assume to be informed by $\overline{k_i}$.

Grounding ourselves in KDD formalisms, a team's knowledge, $\overline{k_i}$, is a precursor to Step 1 in the KDD process, which is to say, $\overline{k_i}$ is a set of prior knowledge on a domain that motivates the capture of data pertaining to that domain. Edges connecting $\overline{k_i}$ to their respective data, $d_i$, in Figure 2.a, can be thought of as a sequence of decisions from steps 2-4 in the KDD processes used to aggregate and refine data. While $\overline{k_i}$ can contain elements of $K^c$, $d_i$ should not be considered as right or wrong as

data will always represent some sub-population of true data, $\hat{D}$, up to some measurement noise $\Delta$. However, since $\overline{k_i}$ instantiates, $d_i$, it is possible that the sub-population $d_i$ may not be suitable to achieve new knowledge in-line with $K$.

## 3.2 The Data Mining Agent

In the mining process (Figure 2.b), *data mining agents* form *data mining teams*, $t_j \subseteq A$. While the agent pool remains the same, we use different team indices to denote the *data mining teams*. In other words, $t_i = t_j$ if and only if $i = j$. Like *experimenting teams*, we assume *data mining teams* have a rectified set of knowledge $\underline{k_j}$. $\underline{k_j}$ is used to inform the data mining approach(s), $F_j$, applied over any raw dataset $d_i$ to extract patterns. We call the set of extracted patterns *information* and denote it $I_{ij}$.

Under the KDD lens, the edges that connect $d_i$ and $\underline{k_j}$ to $I_{ij}$ via $F_j$, in Figure 2.b, can be thought of as a sequence of decisions from steps 5-7 in the KDD process used to extra patterns from data. This includes (5) choosing the function of mining, (6) selecting appropriate algorithms and parameters to produce models, and (7) searching for patterns by leveraging these models, ultimately yielding $I_{ij}$. Not included in the KDD pipeline is the explicit distinction between $\overline{k_i}$ and $\underline{k_j}$. If $i = j$ then crucial team knowledge from the experimenting processes is maintained during the data mining process. This is assumed in the KDD pipeline as it is framed from the perspective of a single agent. However, if $i \neq j$ then $\overline{k_i}$ is potentially lost, resulting in potential misuse or misunderstanding of $d_i$. We argue that this results in an $I_{ij}$ less suitable for achieving new knowledge in-line with the $K$ and potentially subverts the intended use of $d_i$ in regards to $\overline{k_i}$.

## 3.3 The Labeling Agent

In the labeling process (Figure 2.c), *labeling agents* form *labeling teams*, $t_l \subseteq A$. Similar to *data mining teams*, we use different team indices for *labeling agents* to capture distinct teams. That is to say, $t_i = t_j = t_l$ if and only if $i = j = l$. Like the teams that came before, we assume *labeling teams* have a rectified set of knowledge $\breve{k}_l$. The goal of *labeling teams* is to interpret patterns in $I_{ij}$ informed by $\breve{k}_l$ and label those patterns to form a set of new knowledge labels, $k_{ijl}$, where ideally, $k_{ijl} \subset K$, though in practice $k_{ijl}$ likely contains elements in $K$ and $K^c$.

In Figure 2.c, the edges that connect $I_{ij}$ and $\breve{k}_l$ to $k_{ijl}$ capture step 8 in the KDD pipeline, which produces a set of labeled knowledge garnered from $I_{ij}$. Similarly to the data mining step, the KDD pipeline does not formalize instances where knowledge is not propagated forward into the labeling step. If $i = j = l$ knowledge is maintained (again, this is assumed in KDD from the perspective of a single agent). However, if $l \neq j$ then the *labeling team* runs the risk of misusing or misunderstanding $I_{ij}$, unless $t_j$ translated $\overline{k_j}$ onto $t_l$. We argue this potentially results in $k_{ijl}$ containing less elements in $K$ and more elements in $K^c$. This can be further confounded when $j \neq i$ from the mining processes, resulting in a less adequate $I_{ij}$ unless $t_i$ translated their $\underline{k_i}$ onto $t_j$. On the other hand, even if $l = j$ or $t_j$ translated their $\overline{k_j}$ onto $t_l$, if $l \neq i$, the *labeling team* runs the risk of not understanding the initial bias in $d_i$ resulting from $\underline{k_i}$ unless $t_i$ translated their $\underline{k_i}$ onto $t_l$.

## 4 Optimizing an Openness Metric in AI for Science

To demonstrate the utility of MaDiSS, we use our language to construct an argument of knowledge translation among teams asserting that it is imperative teams pass along all provenance required in fulfilling their role in order to optimize a novel openness metric. A crucial question that we have not yet considered is whether an agent or team can truly know if any individual piece of their knowledge is in $K$ or $K^c$. To address this we will make the following assumption.

**Assumption 1** *A human or machine cannot know with exact certainty whether any knowledge element in their knowledge base is a member of $K$ or $K^c$.*

We believe Assumption 1 is justified because no entity has direct access to $K$, and thereby, cannot be fully certain on the membership of any individual piece of knowledge in their knowledge base. Since no upstream team can exactly know if their knowledge is in $K$ or $K^c$, they should always pass

all provenance including accompanying confidence scores, allowing downstream teams to consider their entire context and rectify on their own. The remaining portion of this section will provide a brief monotonicity argument to justify this claim along with potential entry points to aid knowledge translation between teams in line with the AI4OS mission.

Consider an arbitrary $d_i$ constructed by $t_i$ informed by $\overline{k_i}$ as well as a data mining approach(s), $F_j$, of $t_j$ informed by $\underline{k_j}$. From the *data mining team's* perspective, the information, i.e., the set of patterns as defined by a set of chosen languages, yielded by $F_j$, denoted $I_{ij}$, can be defined as the function $F_j : \mathbb{D} \times \mathcal{P}(K \cup K^c) \to \mathcal{P}(\mathbb{P})$ where $\mathbb{D}$ is the set of all possible datasets and $\mathcal{P}(\mathbb{P})$ is the power set of all possible patterns, $\mathbb{P}$, such that

$$I_{ij} = F_j \left( d_i, \underline{k_j} \cup \overline{k_i} \cup \bigcup_{h \in H} \underline{k_h} \right) \tag{1}$$

where $H$ is the set of other *data mining team's* knowledge for which the team $t_j$ may or may not have access. It should also be noted that $t_j$ may or may not have access to $\overline{k_i}$. Similarly, the *labeling team*, $t_l$, also has their interpretation of information, $I_{ijl}$, under their knowledge base and all the other team's knowledge bases of which they have access. This information can be defined as another function $G_l : \mathcal{P}(\mathbb{P}) \times \mathcal{P}(K \cup K^c) \to \mathcal{P}(\mathbb{P})$ such that

$$I_{ijl} = G_l \left( I_{ij}, \check{k}_l \cup \underline{k_j} \cup \overline{k_i} \cup \bigcup_{h \in H} \underline{k_h} \right) \tag{2}$$

where $\underline{k_j}$ may also be empty if inaccessible. If the *data mining* and *labeling team* are the same, $i = j$, then $I_{ij} = I_{ijl}$ because both teams have the same set of combined knowledge. However, if these teams are different, then it may not be the case that $I_{ij} = I_{ijl}$.

Consider a simple example in which the *data mining team*, $t_j$, selects only one data mining approach. In other words, $||I_{ij}|| = ||I_{ijl}|| = 1$. For the sake of this example, let the form of $I_{ij}$ and $I_{ijl}$ be a Knowledge Graph (KG) with nodes representing entities in some system and edges representing the relationships between those entities. For the *data mining team*, $I_{ij}$ is a construction algorithm $F_j$ that learns the edges (relationships) from $d_i$. Now consider the instance in which $\check{k}_l$ contains a piece of knowledge that is absent in $\overline{k_j}$. This could be informed by some specific domain knowledge that states that two potential entities cannot not be directly connected in the KG. If $\overline{k_j}$ is missing this knowledge, and they produce $I_{ij}$ with said edge, the *labeling team* could simply correct the information based on their (assumed to be) more complete knowledge base by removing the erroneous edge. However, this now transforms $I_{ij}$ into the new set of information, $I_{ijl}$.

With the concept of information from the perspective of team $t_l$ in Equation 2, we lastly define a function $L : \mathcal{P}(\mathbb{P}) \to \mathcal{P}(K \cup K^c)$ such that

$$k_{ijl} = L(I_{ijl}, \mathbf{K}^+ \cup \mathbf{K}^-) \tag{3}$$

where

$$\mathbf{K}^+ = \left( \check{k}_l \cup \underline{k_j} \cup \overline{k_i} \cup \bigcup_{h \in H} \underline{k_h} \right) \cap K$$

$$\mathbf{K}^- = \left( \check{k}_l \cup \underline{k_j} \cup \overline{k_i} \cup \bigcup_{h \in H} \underline{k_h} \right) \cap K^c$$

and $k_{ijl} \subset K \cup K^c$ is all the new knowledge translated from the dataset, $d_i$, in accordance with the knowledge and information flow between all the teams in MaDiSS. Equation 3 leads us to the following assumption about $L$ given the following definition of a monotone sequence of sets.

**Definition 1** *Let $X$ be a set and $S \subseteq \mathcal{P}(X)$. A monotone sequence of sets denoted $\langle B_n \rangle_{n \in \mathbb{N}} \in S$ is monotonically increasing if $\forall n \in \mathbb{N} \ B_n \subseteq B_{n+1}$ and is monotonically decreasing if $\forall n \in \mathbb{N} \ B_n \supseteq B_{n+1}$.*

**Assumption 2** *The quantity $\|L(\cdot) \cap K\|$ is monotonically increasing with any monotonically increasing sequence of sets, $\langle K_n^+ \rangle_{n \in \mathbb{N}^+} \subseteq \mathcal{P}(\mathbf{K}^+)$, and the quantity $\|L(\cdot) \cap K^c\|$ is monotonically*

*increasing with any monotonically increasing sequence of sets,* $\langle K_n^- \rangle_{n \in \mathbb{N}^-} \subseteq \mathcal{P}(\mathbf{K}^-)$, *where* $\mathbb{N}^+ = [0, ||K||]$ *and* $\mathbb{N}^- = [0, ||K^c||]$.

In other words, Assumption 2 tells us that the more prior knowledge that is incorporated from all teams in regard to $\mathbf{K}^+$ the greater amount of new knowledge $k_{ijl}$ aligned with $K$ that is produced. Therefore, if ever a new piece of knowledge $k \in K$ is included into $\mathbf{K}^+$, the amount of new knowledge produced that is in $K$ will never decrease. Similarly, the more prior knowledge that is included from all teams in regard to $\mathbf{K}^-$ the greater amount of new knowledge $k_{ijl}$ aligned with $K^c$ that is produced. We believe this is a justified assumption as included knowledge in $K$ would never lead to a knowledge element $K^c$ and vice versa.

Therefore, if all knowledge is equally important to discover, a natural metric to gauge the success of a knowledge translation is to measure the quantity:

$$\left\| \bigcup_{i,j,l} k_{ijl} \cap K \right\| - \left\| \bigcup_{i,j,l} k_{ijl} \cap K^c \right\| \tag{4}$$

The goal of knowledge translation, and by extension, a MaDiSS and AI4OS, is to maximize Equation 4. However, recall that Assumption 1 states that the membership of elements in $k_{ijl}$ is unknowable, so Equation 4 is not directly computable. Therefore, it is the job of a MaDiSS to approximate $K$, and by extension $K^c$. In fact, each team's knowledge resembles their best approximation of a subset of $K$ already. However, on receiving translated knowledge from upstream, we conclude that a successful MaDiSS should help process, filter, reframe, and/or reweight this knowledge to increase the expectation that the amount of knowledge in $K^+$ is maximized and the amount in $K^-$ is minimized. We argue this will maximize Equation 4 under Assumption 2. Lastly, two extensions to Equation 4 could be to normalize the metric with respect to the cardinality of $k_{ijl}$ in order to compare across domains as well as relax our assumption that all knowledge is equally important.

## 5 Why Openness in AI for Science

The pursuit of openness in AI4OS can be viewed as an effort to maximize knowledge dissemination, achieved through the effective conveyance of essential data and algorithmic provenance to the team responsible for labeling knowledge as well as its intermediaries (Equation 4). Further, the optimization of this process, which can better align generated knowledge with ground truth, can only be achieved by open knowledge sharing among collaborators. Therefore, we assert that the primary ethical concerns in AI4Science revolve around open access to and open communication of AI-derived knowledge discoveries.

Broad communication of knowledge discoveries is not a new ethical consideration and has been handled extensively by the Open Science community (18; 19; 20; 21; 22). However, a critical facet of analyzing the ethics of these systems involves examining their increased speed and autonomy. If these systems become entirely self-driven, the rapid acceleration of discovery may overwhelm our society's capacity to absorb it, as exemplified by the challenge of digesting the vast research output from predatory journals and conferences (29; 30). Accelerating knowledge discovery through AI4Science without due consideration could further burden researchers and hinder scholarship.

Another ethical concern is the risk of restricting open access to AI-derived scientific discoveries by industry, governments, and the military. These entities may attempt to silo AI4Science implementations for their benefit, limiting the knowledge available for labeling exclusively to their organizational unit. To uphold operationally moral (31) standards, society should ensure diverse agent participation in knowledge translation teams and agents through increased regulation, auditing, and disclosure (32). Given the current self-regulatory environment of most AI companies (33), AI ethicists must continue to press for specific enforceable and non-voluntary legislation (34; 35) of AI4Science systems focusing their accessibility, transparency, and openness.

Additionally, we must consider how denial of access could exacerbate existing societal inequities via socio-technical forces. For instance, if new knowledge generated by a DiSS is used to accelerate individual knowledge acquisition rather than novel scientific discovery, denying access to such systems based on social class or identity could further marginalize underprivileged populations,

perpetuating systemic disparities. For example, consider an AI-driven system that aids in childhood education by increasing the speed at which a child acquires new knowledge. Then denial of access based on class or identity could further limit a marginalized child's performance. This effect would fortify and continually widen systemic inequities among historically oppressed communities.

Shifting our focus to specific ethical considerations within the MaDiSS formulation, we emphasize the importance of the *experimenting team* documenting and transparently communicating assumptions, biases, and contextual factors affecting dataset collection, processing, or experimentation. This information is crucial for both the knowledge labeling process and data mining. Neglecting to provide this information may lead to mislabeled knowledge and exacerbate ethical concerns. While initiatives like "datasheets for datasets" are steps in the right direction (36), a robust provenance architecture for communicating this knowledge remains an open challenge.

The significance of Open Data initiatives (23; 24; 25; 26) and the role of provenance in databases (27; 28) has been extensively studied, particularly within the realm of biomedical data. However, the adoption of these practices within the machine learning community as part of the knowledge translation process has been notably sluggish and is poised to face further challenges with the rise of AI-driven systems. An associated area that has received inadequate attention, evident from a MaDiSS perspective, is the imperative to communicate data mining or algorithmic provenance to upstream consumers. The current focus in this domain has primarily centered on the reproducibility of machine learning algorithms (37; 38). However, MaDiSS underscores the need to go beyond reproducibility and ensure that downstream labeling teams consider the underlying algorithmic assumptions, biases, and contexts. Building on the framework proposed by Gebru et al., this could be implemented through an "information sheet for information systems" approach, fostering community collaboration to identify intents, assumptions, and contexts that must be captured to enhance the successful knowledge translation process and supports the adoption of AI4OS.

Our final ethical consideration pertains to the role of the labeling team in acknowledging and considering the provenance provided by the experimenting and data mining teams. This aspect relies on human autonomy, as knowledge labelers can choose to ignore provenance. As human autonomy in the knowledge translation process becomes curtailed by AI4Science manifestations like self-driving labs, this becomes increasingly problematic. Consequently, fostering ethical work cultures becomes imperative, wherein all providential knowledge is not only valued but also demanded from downstream agents. A pragmatic societal solution involves enhancing digital literacy among labeling teams, emphasizing the significance of these communication channels, and aligning with the contextualist recommendations for AI use as presented by Chan (32).

Nevertheless, this requirement to thoughtfully consider provenance encounters socio-technical resistance. AI professionals and researchers who currently leverage DiSS and soon AI4Science systems inevitably contend with external pressures that may hinder their consideration of information or data context. These pressures could stem from impending deadlines, managerial demands, or market forces. Therefore, it becomes crucial for researchers to prioritize knowledge transfer, ensuring that their respective labels are well-informed to avert potentially detrimental consequences. Most significantly, this underscores a concluding ethical consideration directed at the developers of AI4Science.

## 6    Conclusions and Future Work

In this paper, we introduced AI for Open Science using a Multi-agent Discovery Support formalism that aligns with Open Science ideals and highlights the need for openness within the AI for Science community. We demonstrated the power of this formalism to make assertions that appropriate labeling of new knowledge is monotonically related to the degree to which the labeling agent understands the intent, context, and assumptions of the upstream teams. This imposes a clear ethic on all teams operating within an AI for Open Science paradigm to propagate their prior knowledge to all downstream teams. Finally, we assessed ethical considerations of openness in the context of AI for Science justifying the need for systems to adopt an AI for Open Science framing.

Additionally, we believe there are several fruitful directions for future work. First, Assumption 2 must be empirically validated, perhaps through human team-based experimentation. Second, we need to assess as a field the degree to which current AI-driven systems are effectively closed or open. However, for this to be accomplished we must develop a proxy for Equation 4 that is computable and provably optimizes our openness metric.

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
