# OpenReview forum: "AI for Open Science: A Multi-Agent Perspective for Ethically Translating Data to Knowledge"
_NeurIPS.cc/2023/Workshop/AI4Science — NeurIPS2023-AI4Science Poster_

### Official Review · Reviewer_d7Jd · 2023-10-23
**An interesting but hard-to-envision vision**

**Rating:** 6
**Confidence:** 4

**Review:**

Quality:

-	Strengths: The authors describe a concept where AI systems share information openly to maximize the dissemination of scientific knowledge. Their concept is thoroughly explained.
-	Weaknesses: The authors’ concept requires many unrealistic conditions, and may be too optimistic of human behavior to ever be implemented or effective. Along this line, the authors also make strong assumptions about the “inevitable” future of AI in replacing most scientific pursuits by mankind, which may not be “becoming increasingly clear” to some readers. Overall, although a neat picture, the concept needs to be more grounded in reality.

Clarity:

-	Strengths: The paper is well-structured and understandable for scientists. It provides a clear context and method explanation, and is well cited.
-	Weaknesses: The authors' concept is lacking a pithy statement that explains their entire vision. Currently, there seem to be two definitions of openness: (1) the authors seem to envision a network of AI models remotely performing different scientific tasks but interlinked so as to most efficiently share discoveries as they occur in isolated contexts; and (2) the discoveries made by these open-sharing networks are then disseminated to all mankind without discrimination. Thus, there is a difference between openness among AI agents and among AI agents and humans that should be addressed early on to avoid ambiguity.
-	As well, it is not clear how AI agents are created in this futuristic world. Are they designed from identical template(s) – architectures, training protocols, and objective functions – to enforce their ethical alignment? Or will it be allowed that privately designed agents can contribute openly to other (possibly-public or privately owned) agents. Either approach expands into pros/cons as detailed below (Significance).

Originality:

-	Strengths: The authors claim to modify established principles to create a superset of principles for establishing AI applications in scientific discovery.
-	Weaknesses: This reviewer is not equipped to assess the novelty of the work.

Significance:

-	Strengths: In an ideal and simple world without competing interests among different groups of humans, with AI models capable of reaching or exceeding human performance, and doing so without error in AI results, the proposed principles should work to optimize scientific discovery.
-	Weaknesses: However, the world is far from ideal. For example, if privately-owned AI agents are allowed on the network, it will boost competition in designing better AI agents, but may prove harder to collectively guide these models towards ethical results. If private AI agents are not allowed, there will be no incentive towards innovation by mankind of the envisioned AI hivemind. In the prior scenario, how then will diverse agent participation be regulated, especially if there is no centralized body or government enforcement? Regardless, the authors optimistically assume that the findings by AI will be accurate, but it is a far greater concern than one of openness if AI agents learn how to take shortcuts (cheat) to optimize their publication objective. Surely brilliant AI agents will learn to outsmart even the most cunning human reviewers assessed with catching them. The dissemination of incorrect information by such AI is far more detrimental than no dissemination, and may not become apparent for many generations of models. If anything, it is more obvious that AI results should be contained from contributing to unchecked contamination of the compendium of scientific knowledge, and thoroughly vetted before synthesis by humans into a collective whole.
-	There is a related problem to consider when considering “model collapse”. Already, researchers have found that as generations of AI models progress on their previously-generated outputs, models start become increasingly problematic. As of now, it appears like generative AI models must train on human-produced data to function. For example, imagine a world of AI agents which are trained largely on databases such as Wikipedia (as they are currently). If these AI agents are allowed to make revisions and add content (AI-generated news or scientific discoveries) to Wikipedia over many generations, eventually there will be agents that train on such a degenerate version of Wikipedia that their predictive performance completely collapses. Expanding this vision to the reach of the authors' paper, such should also be the case for AI agents updating and training over generations on the compendium of AI-generated scientific knowledge.

---

### Official Review · Reviewer_rUPj · 2023-10-23
**A nice theoretical framework which brings open science to ai4science. However, no concrete steps are proposed to fix ethical issues.**

**Rating:** 6
**Confidence:** 2

**Review:**

I found section 2.2 to be informative and provide nice background.

I would encourage the authors to provide suggest how real-world implementations of this idea might work in follow-up work.


I appreciate that the ethical arguments made were mathematically grounded in the KDD process, at least to an extent.

I would suggest the authors to integrate the idea of a feedback loop more clearly earlier on. In particular, this can be added to figure 2. Since the amount of available knowledge is vast, it is important to use new knowledge (2.c.) to inform knowledge collection. Further, using uncertainty quantification from patterns mined in 2.b. to direct the data creation process would also be useful.

For definition 1, should it be monotonically nondecreasing and monotonically nonincreasing? since an improper subset was used which allows equality. Does this pose problems for the scientific process? Can agents become stuck somehow in a position where no new knowledge is being discovered?


It would be good to try and quantify the affect of some percent of "bad agents" on the relative rates of knowledge growth in K vs K^c. This already happens in this framework if we ignore Ai4Science and only consider human teams.

"We believe this is a justified assumption as included knowledge in K would never lead to a knowledge element Kc and vice versa."
-- a agree this makes sense, but is it possible to show there are no such edge cases? Are there any nonlinear relations between knowledge from K and Kc which cause disproportionate growth in Kc?



Overall, I liked the paper and I think it brings up an important point of discussion which is quite relevant to this workshop. I am less familiar with the ethics side of machine learning. Subjectively, some portion of the ethics community isn't grounded in ethics, but rather contrarianism and emotion. Having said that, I quite like how this work grounds into the KDD process. My biggest personal issue here, which I sincerely hope the authors will pursue in follow-up work, is that the conclusions have little practical meaning at present. I agree it is best to maximize the discovery of "true" knowledge (whatever that means) over false knowledge. But, how can we actually do this? How do bad agents impact the process? Would the authors be able to create a software implementation on a specific ML problem that researchers can then operate within? I would have liked to see more discussion of the real-world possibilities of this framework instead of Section 5. Section 5 takes up a lot of space and I don't think most researchers actually need much convincing about the merits of open science. The paragraphs make sense, but I would prefer more discussion of how to fix these issues and less discussion of the issues, since I imagine that is discussed elsewhere.

---

### Meta-Review · Area_Chair_JvQW · 2023-10-26

**Recommendation:** Accept (Poster)
**Confidence:** 5

**Metareview:**

Overall the reviewers are in agreement that the work provides an original and interesting perspective, but that it requires further refinement and consideration. The  overall strengths and limitations are as follows:

Strengths:
- MaDiSS system is interesting, and the ways it takes ethical concerns into consideration
- Paper is clear and easy to follow

Limitations:
- Lacks of concrete and concise articulation of AI models/agents in scenarios considered
- Improving structure and flow (e.g., earlier introduction of feedback loop)
- Provide more grounded real world examples, especially clarifying definitions around true and false knowledge.
- Some of the claims are difficult to justify (e.g., modify established principles of scientific discovery)
- Requires deeper considerations of  pertinent edge cases, model collapse

The authors may wish to closely consider the listed limitations in preparing this work for its next steps. As a note: I also found the approach that the authors take interesting, but I would have liked to see a bit more consideration around the effects of agent composition in teams. AI agents work in ways that humans can't (e.g., without eating or sleeping) and humans can create some datasets that AI agents (currently) can't (e.g., collect biological specimens and extracting their DNA). So what happens when the teams are imbalance or in the extreme contain only one type of agent? From the HCI literature there are some interesting (qualitative) studies that describe H-AI teaming dynamics  in data work (KDD) and its curious how that could be model by MaDiSS.